## RESEARCH ARTICLE

# Nuclear enlargement induced by overexpression of nuclear export signal is associated with abnormal nuclear division in *Schizosaccharomyces pombe*

Takahiro Fujimoto[1], Suzu Watanabe[2], Yuko Imamura[2], Masaki Mizunuma[1] and Kazunori Kume[1,2,*]

## ABSTRACT

The size of the nucleus is tightly coordinated with cell size across eukaryotes, yet the physiological significance of maintaining proper nuclear dimensions remains poorly understood. Here, we investigate how nuclear size dysregulation resulting from perturbed nucleocytoplasmic transport affects mitotic fidelity in *Schizosaccharomyces pombe*. Overexpression of a GFP-tagged nuclear export signal (NES-GFP) induced nuclear expansion, leading to severe growth defects and frequent errors in chromosome segregation during mitosis. Live-cell imaging revealed that enlarged nuclei underwent delayed mitotic progression and abnormal nuclear division. Strikingly, genetic suppression of nuclear expansion alleviated these defects, whereas enhancement of nuclear size exacerbated them. Together, these findings suggest that maintaining proper nuclear dimensions contributes to accurate chromosome segregation, although additional effects of NES-GFP overproduction and other factors influencing nuclear size should be further examined.

KEY WORDS: Nuclear size, Fission yeast, Closed mitosis, Nucleocytoplasmic transport

## INTRODUCTION

Nuclear size is a highly regulated and conserved feature of eukaryotic cells. In most organisms, the nuclear volume scales with cell size, maintaining a constant nuclear-to-cytoplasmic (N/C) ratio – a cellular phenomenon historically known as the karyoplasmic ratio (Hertwig, 1903; Boveri, 1905). This size scaling is thought to be important for numerous nuclear functions, including chromatin organization, RNA processing, and gene expression (Edens et al., 2013; Cantwell and Nurse, 2019b). Although the molecular mechanisms contributing to nuclear size regulation have been gradually uncovered (Jorgensen et al., 2007; Neumann and Nurse, 2007; Levy and Heald, 2010; Hara and Merten, 2015; Kume et al., 2017, 2019; Kume, 2020; Cantwell and Nurse, 2019a,b; Jevtic et al., 2019; Lemière et al., 2022), the functional importance of maintaining an appropriate nuclear size remains largely unclear.

[1]Program of Biotechnology, Graduate School of Integrated Sciences for Life, Hiroshima University. [2]Program of Biomedical Science, Graduate School of Integrated Sciences for Life, Hiroshima University, 1-3-1 Kagamiyma, Higashi-Hiroshima, Hiroshima 739-8530, Japan.

*Author for correspondence (kume513@hiroshima-u.ac.jp)

 K.K., 0000-0002-0613-8081

Organelles such as mitochondria, vacuoles, and the mitotic spindle also scale with cell size, and their sizes affect specific cellular processes. For example, spindle length and assembly kinetics scale with cell volume to ensure accurate chromosome segregation (Hazel et al., 2013; Rieckhoff et al., 2020), and the efficacy of the spindle assembly checkpoint is influenced by cell size (Galli and Morgan, 2016). Similarly, vacuolar and mitochondrial dimensions affect pH homeostasis, energy production, and inheritance (Chan and Marshall, 2010, 2014; Miettinen and Björlund, 2017). These findings suggest that appropriate size control of organelles is not merely a structural constraint, but a functional necessity.

Despite this, the direct physiological impact of nuclear size dysregulation remains insufficiently understood. Recently, we found that overexpression of a GFP-fused nuclear export signal (NES-GFP) in *Schizosaccharomyces pombe* disrupts nucleocytoplasmic transport. The NES sequence, derived from protein kinase A inhibitor, binds tightly to Exportin in human cells, thereby interfering with nuclear export (Engelsma et al., 2004; Güttler et al., 2010; Fu et al., 2018). This disruption causes nuclear accumulation of cargo proteins and the formation of intranuclear microtubule bundles, leading to an increase in nuclear size (T. Fujimoto, unpublished observations; manuscript under revision). Although it has been shown that highly expressed NES-GFP causes severe growth defects in *Saccharomyces cerevisiae* (Kintaka et al., 2016), whether increased nuclear size induced by NES-GFP overexpression affects cellular fitness or specific cell cycle events in fission yeast remains unexplored.

In this study, we characterize the effects of NES-GFP overexpression on cell growth in *S. pombe*. We show that nuclear enlargement associated with the formation of intranuclear microtubule bundles under NES-GFP overexpression causes chromosome segregation defects during mitosis. Furthermore, we demonstrate that genetic suppression or enhancement of this nuclear enlargement modulates the severity of these defects. These findings highlight the importance of nuclear size homeostasis in maintaining genome stability.

## RESULTS AND DISCUSSION

### NES-GFP overexpression compromises cell proliferation and mitotic fidelity

To determine the effect of NES-GFP overexpression on cell proliferation in fission yeast, we performed growth assays using wild-type (WT) cells carrying NES-GFP or GFP under the thiamine-repressible *nmt1* promoter in pREP1 plasmid (Maundrell, 1994). WT cells overexpressing GFP or NES-GFP grew normally on EMM agar plates containing thiamine at 28°C. In contrast, cells overexpressing NES-GFP failed to grow on EMM plates without thiamine (Fig. 1A), indicating that NES-GFP overexpression impairs cell proliferation in *S. pombe*. We next monitored growth in EMM liquid medium, in which the induction of NES-GFP overexpression under *nmt1* promoter occurs at around 16 h (Maundrell, 1994). As shown in

Biology Open

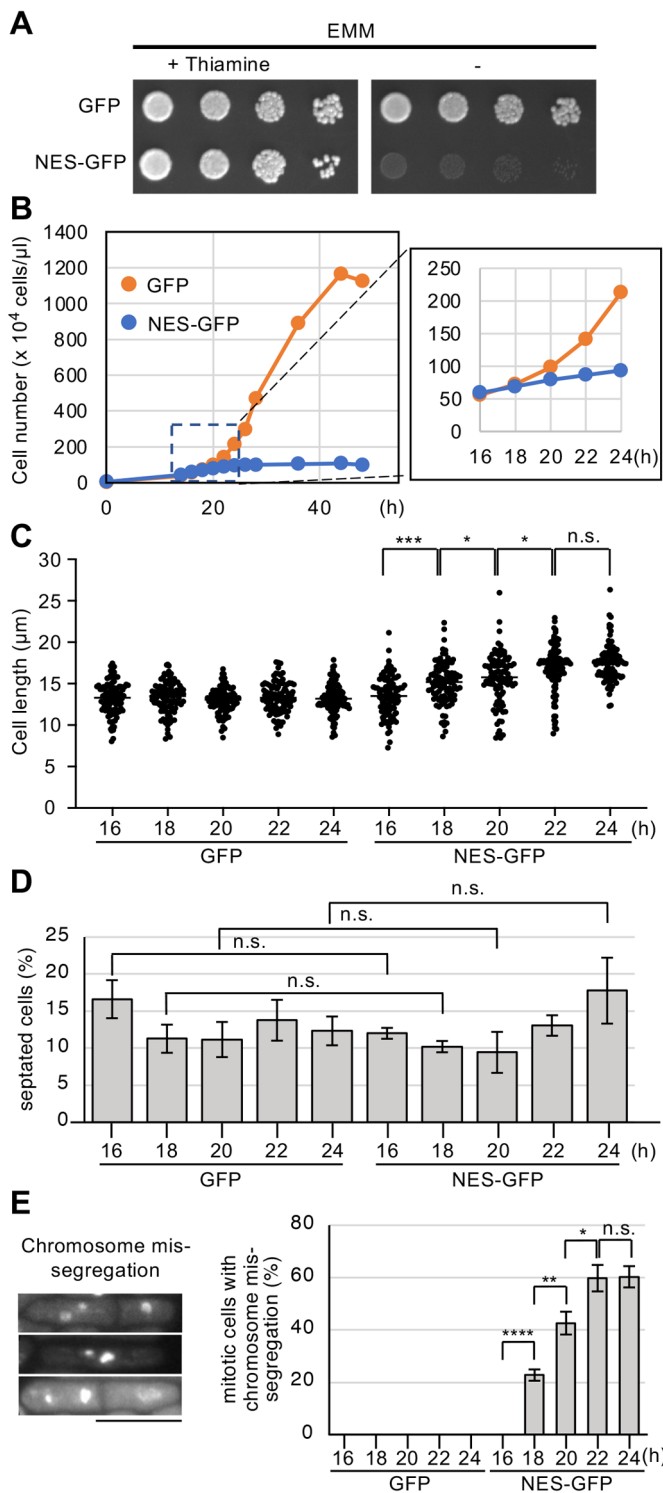

**Fig. 1. Effect of NES-GFP overexpression on cell growth in fission yeast.**
(A) Growth assay on EMM agar plates with or without thiamine at 28°C. Thiamine represses gene expression downstream of the *nmt1* promoter. Cells overexpressing GFP or NES-GFP were grown in minimum medium at 28°C and approximately 5 μl of 5×10⁵ cells/μl spotted on the plates, with the highest concentration on the left and sequential fivefold dilutions towards the right. (B) Growth curve of cells overexpressing GFP or NES-GFP grown in EMM liquid medium at 28°C. (C) Cell length of septated cells overexpressing GFP or NES-GFP at indicated time points. Cells were grown in EMM medium at 28°C (*n*>50 cells per experiment; three independent experiments). Statistical significance was assessed using two-tailed Welch's *t*-test. *P<0.05, ***P<0.001. n.s., not significant. (D) The frequency of septated cells overexpressing GFP or NES-GFP at indicated time points. Cells were grown in EMM medium at 28°C (*n*>200 cells per experiment; three independent experiments). Statistical significance was evaluated using the two-tailed nonparametric Mann–Whitney *U* test. (E) (left) Images of cells with chromosome mis-segregation observed when NES-GFP was overexpressed. DNA was stained with DAPI. (right) The frequency of mitotic cells with chromosome mis-segregation at indicated time points. Cells overexpressing GFP or NES-GFP were grown in EMM medium at 28°C (*n*>200 cells per experiment; three independent experiments). Statistical significance was assessed using Chi-square test. *P<0.05, **P<0.01, ****P<0.0001. n.s., not significant.

observed chromosome segregation by staining DNA with 4,6-diamidino-2-phenylindole dihydrochloride (DAPI). Compared to GFP overexpressing cells, NES-GFP overexpressing cells exhibited a substantial increase in unequal chromosome segregation at 18 h after the NES-GFP induction (Fig. 1E). The frequency of chromosome segregation defects increased approximately threefold at 24 h (Fig. 1E). Collectively, these findings demonstrate that NES-GFP overexpression impairs cell proliferation and increases the frequency of chromosome segregation errors during mitosis in *S. pombe*.

## Nuclear enlargement delays mitotic spindle formation and disrupts chromosome segregation

To investigate the mechanisms of chromosome mis-segregation in cells overexpressing NES-GFP, we performed live-cell imaging of cells co-expressing nuclear envelope and microtubule markers (Cut11-mCherry and Atb2-mCherry). During interphase, cells overexpressing NES-GFP exhibited an enlarged nucleus with intranuclear microtubule bundles (Fig. S1). The formation of intranuclear microtubule bundles and nuclear enlargement were observed after 18 h and became more pronounced at 20 h (Fig. S1, Table S1). Importantly, the interphase intranuclear microtubule bundles disappeared prior to the formation of spindle microtubules upon entering mitosis (Fig. 2A). Indeed, interphase intranuclear microtubule bundles that protruded from the nucleus did not coexist with the mitotic spindle.

Upon mitotic entry, cells overexpressing NES-GFP displayed delayed spindle formation, compared with that of GFP overexpressing cells (Fig. 2B,C,D). Quantitative analysis revealed that, at 20 h after the induction of NES-GFP overexpression, 42% of cells showed abnormal nuclear divisions with delayed spindle elongation (Fig. 2E). The delayed spindle elongation resulted in unequal nuclear division, producing large and small daughter nuclei in 23% of mitotic cells (Fig. 2C; hereafter referred to as unequal nuclear division). In contrast, 19% of cells underwent nuclear division without a spindle, occurring in the direction opposite to spindle elongation, and subsequently completed division by spindle formation after septation (Fig. 2D; hereafter referred to as spindle-independent nuclear division). DNA staining with Hoechst 33342 confirmed the presence of chromosomes in both nuclei during the division without a spindle (Fig. 2F), indicating that nuclear division occurred independently of canonical spindle-mediated nuclear division.

Fig. 1B, the number of NES-GFP overexpressing cells remained low with slight increase after 18 h, whereas GFP overexpressing cells increased and reached a stationary phase. To explore the cause of growth defects, we measured cell length and the frequency of septated cells. The cell size of NES-GFP overexpressing cells gradually increased after 18 h, reaching an average length of 18 μm at 24 h, 1.4 times larger than that of GFP overexpressing cells (Fig. 1C). This increase in cell size appeared to be accompanied by a delay in the cell cycle, but the proportion of septated cells increased after 22 h, indicating that cell division had proceeded (Fig. 1D). We next

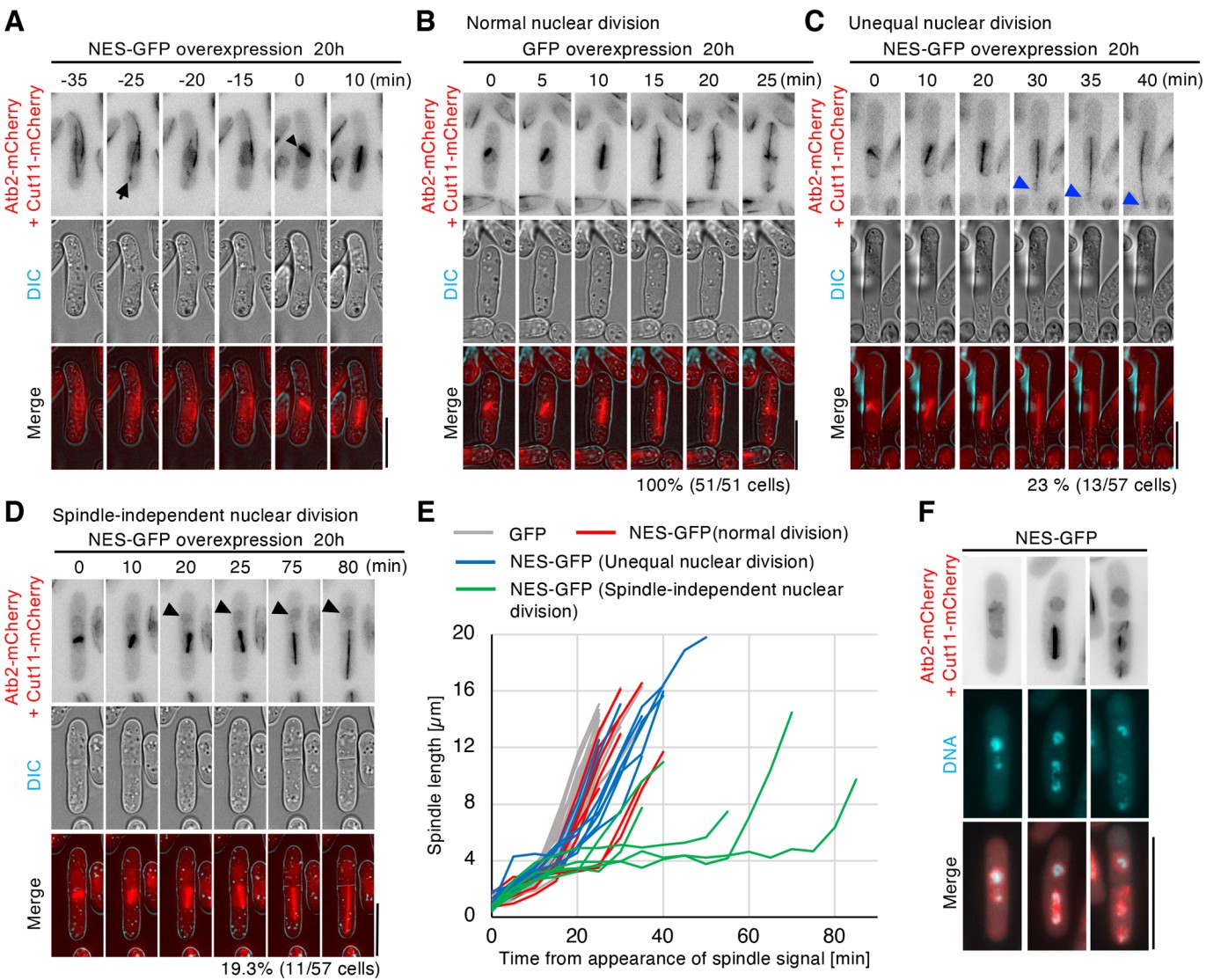

**Fig. 2. The overexpression of NES-GFP causes nuclear division defects through delayed spindle elongation.** (A-D) Representative time-lapse images with 5 min intervals of cells overexpressing GFP or NES-GFP. Cut11-mCherry and Atb2-mCherry (red), DIC (gray). Scale bars: 5 μm. (A) Time-lapse images shown spindle (arrowhead) formation after diminished intranuclear microtubule bundles (arrow) in cells overexpressing NES-GFP. (B) Nuclear division of a cell overexpressing GFP. All cells divided normally (51 cells). (C,D) Abnormal nuclear divisions in cells overexpressing NES-GFP (57 cells). Blue and black arrowheads indicate a small, divided nucleus and a nucleus divided without a spindle, respectively. (E) Spindle lengths of cells overexpressing GFP or NES-GFP were plotted. Time 0 indicates the onset of spindle formation, and measurements were taken until the spindle disappeared. (F) Representative snapshot images of spindle-independent nuclear division observed in cells overexpressing NES-GFP. Cut11-mCherry and Atb2-mCherry are shown in red, and DNA (Hoechst) is shown in cyan. Scale bars: 10 μm. Time-lapse imaging was performed in three independent experiments, and more than 50 cells were analyzed.

Intriguingly, a correlation was observed between the pre-division N/C ratio and mitotic defects. NES-GFP overexpressing cells that divided normally had an average pre-division N/C ratio of 0.096, whereas cells with abnormal nuclear divisions had an average N/C ratio of 0.11 with delayed spindle elongation (Fig. 3A,B, Table S2). Our observations indicate that spindle-independent nuclear division occurs mainly in cells with a pre-division N/C ratio above 0.11 (Fig. 3A,B, Table S2), suggesting that excessive nuclear enlargement impairs spindle formation and chromosome segregation.

Notably, the occurrence of spindle-independent nuclear division in cells overexpressing NES-GFP is consistent with previous observations in *S. pombe* treated with microtubule polymerization inhibitor carbendazim (Castagnetti et al., 2010). This similarity raises the possibility that nuclear enlargement triggers an alternative

division pathway that bypasses canonical spindle-mediated segregation when spindle assembly is compromised.

These observations indicate that nuclear size is not merely a passive structural feature but actively contributes to coordinating mitotic architecture. The delayed transition from interphase intranuclear microtubule bundles to spindle formation may represent a mechanistic link between nuclear enlargement and altered mitotic timing. Collectively, our data support the idea that maintaining an appropriate nuclear volume relative to cell size is crucial for faithful mitotic progression and chromosome segregation.

## Genetic modulation of nuclear size alters the severity of mitotic phenotypes

To test whether nuclear size per se influences the fidelity of mitosis, we used genetic mutants that either suppress or enhance nuclear

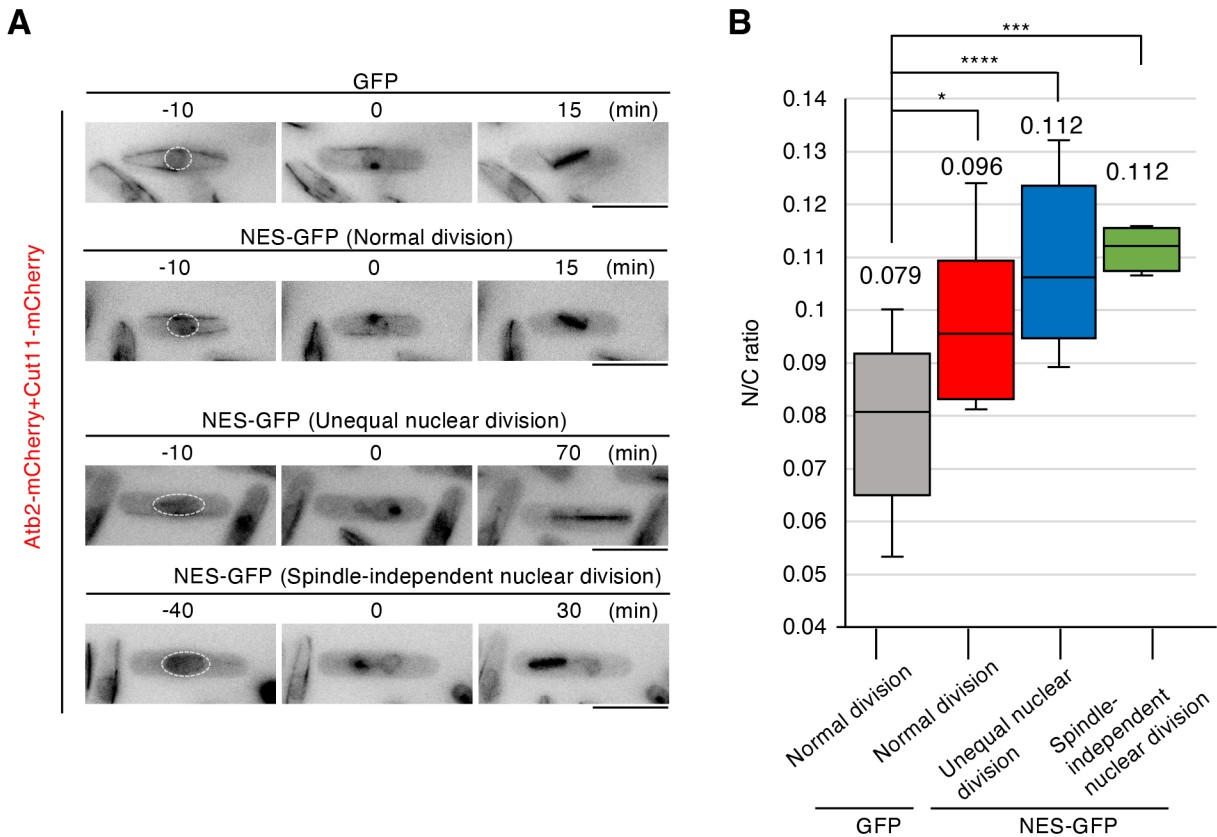

**Fig. 3. Pre-division N/C ratios of normal and abnormal nuclear divisions observed in NES-GFP overexpressing cells.** (A) Time lapse images of cells overexpressing GFP or NES-GFP. Cut11-mCherry and Atb2-mCherry (red). Dotted lines indicate interphase nucleus. (B) Box-and-whisker diagram showing pre-division N/C ratio. GFP (*n*=20), NES-GFP (normal division, *n*=7), NES-GFP (NES-GFP (abnormal nuclear division shown in Fig. 2C, *n*=9), NES-GFP (abnormal nuclear division shown in Fig. 2D, *n*=5). Values indicate the average N/C ratios. Statistical significance was assessed using two-tailed Welch's *t*-test. * *P*<0.05, ****P*<0.001, *****P*<0.0001. Scale bars: 10 µm.

expansion in cells overexpressing NES-GFP. Deletion of *imp1*, one of the importin-alpha genes in *S. pombe* (Umeda et al., 2005), attenuated the formation of intranuclear microtubule bundles induced by NES-GFP overexpression and suppressed the increase in the N/C ratio. The mean N/C ratios in *imp1Δ* cells overexpressing NES-GFP at 18 and 20 h were 0.089 and 0.095, respectively, compared with 0.106 and 0.12 in WT cells overexpressing NES-GFP (Fig. S1, Table S1), as also observed in our recent study (T. Fujimoto, unpublished observations; manuscript under revision). Correspondingly, the nuclear division defects shown in Fig. 2C and D were alleviated in *imp1Δ* cells overexpressing NES-GFP (Fig. 4A,B). Unequal nuclear division was absent at 18 h and observed in 10% of cells at 20 h in *imp1Δ* cells overexpressing NES-GFP, compared with 19% and 20% in WT cells overexpressing NES-GFP, respectively. Spindle-independent nuclear division was also absent at 18 h and observed in 5% in *imp1Δ* cells overexpressing NES-GFP, compared with 9% and 20% in WT cells overexpressing NES-GFP, respectively (Fig. 4B). These findings indicate that the suppression of nuclear expansion-where the ratio was maintained below 0.1-mitigates the severity of mitotic abnormalities, supporting the view that nuclear size contributes to accurate chromosome segregation.

Conversely, deletion of *mto2*, a γ-tubulin complex linker gene required for cytoplasmic microtubule nucleation (Borek et al., 2015; Kume et al., 2024), promoted the formation of intranuclear microtubule bundles and further enhanced nuclear expansion at 16 h upon NES-GFP overexpression (Fig. S1). At 16 h, the frequency of intranuclear microtubule formation in *mto2Δ* cells overexpressing

NES-GFP was approximately 50% and reached a plateau at 18 h. The N/C ratio in *mto2Δ* cells overexpressing NES-GFP at 16 h, but not at 18 or 20 h, was significantly higher than that of WT cells overexpressing NES-GFP (Fig. S1, Table S1). Consistently, nuclear division defects were observed in *mto2Δ* cells overexpressing NES-GFP at 16 h, but not in WT cells overexpressing NES-GFP (Fig. 4A,B). Furthermore, the frequency of cells exhibiting spindle-independent nuclear division was gradually increased in both WT and *mto2Δ* cells overexpressing NES-GFP, as the N/C ratio increased (Fig. 4, Fig. S1), suggesting that spindle-independent nuclear division is likely caused by nuclear enlargement. Taken together, these results indicate that nuclear size is an important factor influencing mitotic fidelity in closed mitosis. Genetic modulation of nuclear size correlates with changes in the accuracy of chromosome segregation, highlighting the importance of maintaining nuclear size homeostasis. However, we cannot exclude the possibility that the formation of intranuclear microtubule bundles induced by NES-GFP overexpression may also affect mitotic fidelity, and further analyses are required to clarify this point.

Our study demonstrates that proper nuclear size contributes to the fidelity of mitosis in *S. pombe*, which undergoes closed mitosis. Overexpression of NES-GFP induced nuclear expansion, disrupting the coordination of mitotic progression and leading to chromosome segregation errors. Genetic modulation of nuclear size through *imp1* and *mto2* deletions revealed that suppression of the nuclear expansion alleviates mitotic defects, whereas enhancement of nuclear size exacerbates them. These observations support a causal

Biology Open

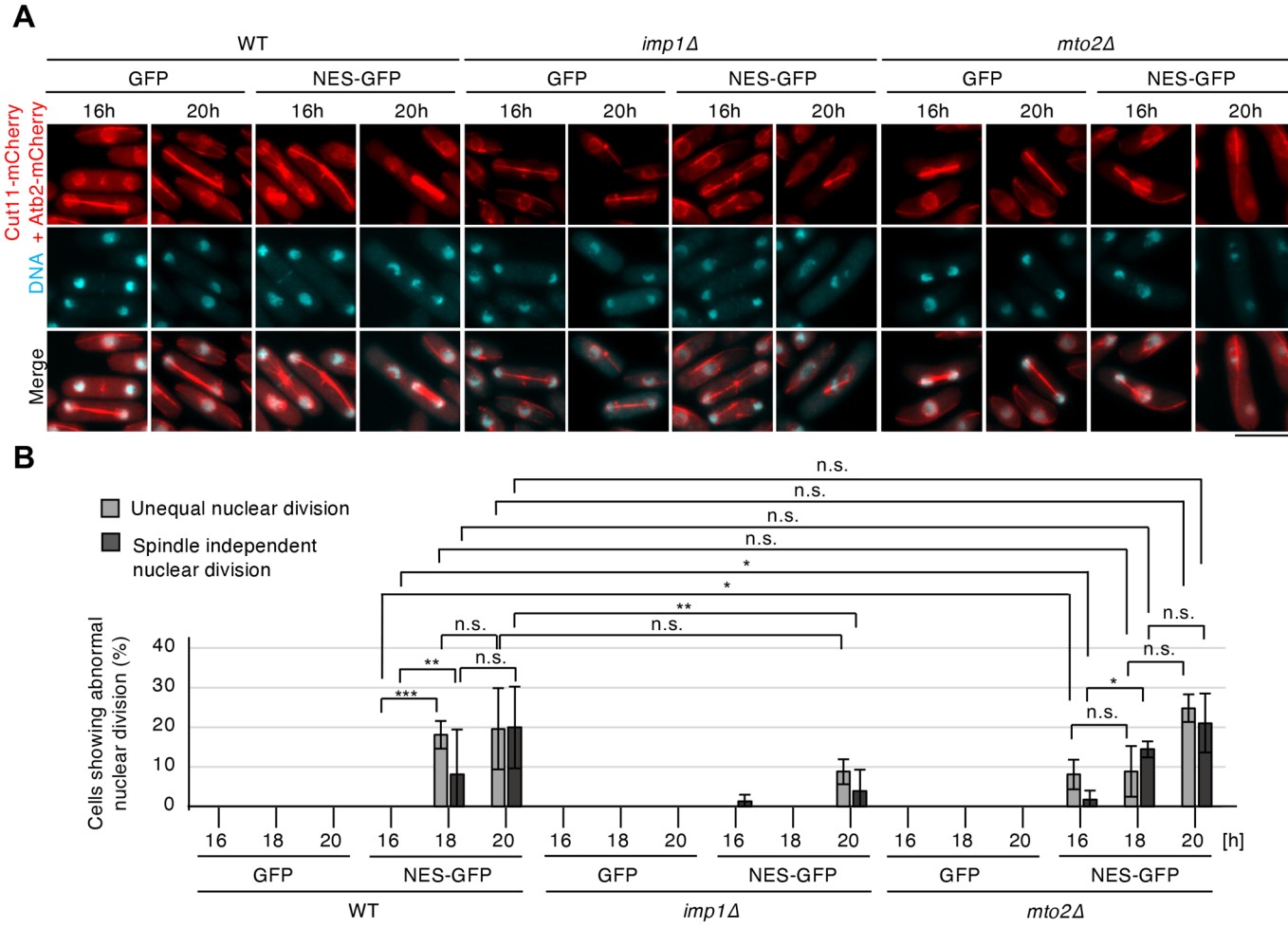

**Fig. 4. Chromosome mis-segregation are induced in correlation with an increase in the N/C ratio due to overexpression of NES-GFP.** (A) Images of cells overexpressing GFP or NES-GFP. Cut11-mCherry and Atb2-mCherry (red), and DNA (Hoechst) (cyan). (B) Frequencies of cells showing unequal nuclear division or spindle-independent nuclear division among cells exhibiting normal or abnormal nuclear division (*n*>200 cells per experiment; three independent experiments). Statistical significance was assessed using the Chi-square test. *$P<0.05$, ***$P<0.001$, ****$P<0.0001$. n.s., not significant. Error bars: s.d. Scale bar: 10 µm.

relationship between nuclear size and the accuracy of chromosome segregation during closed mitosis. Given the limitations of this study, future investigations using other nuclear size mutants (Kume et al., 2017; Helena and Nurse, 2019a) will be essential to confirm and extend these conclusions.

Nuclear expansion is a hallmark of cancer and aging cells (Zink et al., 2004; Webster et al., 2009; Edens et al., 2013), where genome instability frequently occurs. Our results suggest that perturbation of nuclear size homeostasis may represent a conserved risk factor for defective chromosome segregation across eukaryotes. Thus, nuclear size should be considered not only a structural trait but also a key regulatory parameter of genome stability and cell proliferation. Future studies should address the molecular pathways that sense and control nuclear size, and how these pathways intersect with the spindle assembly and chromosome organization. While the mechanisms may differ between organisms undergoing closed mitosis versus open mitosis, nuclear size abnormalities are broadly associated with genome instabilities, particularly in cancer and aging cells. Thus, our work provides a conceptual framework for considering nuclear size homeostasis as a potential risk factor for genome instability and highlights the importance of investigating how nuclear geometry contributes to genome stability across diverse eukaryotes.

## MATERIALS AND METHODS
### Strains, growth conditions and plasmids
Yeast strains, media and general methods were as described previously (Moreno et al., 1991). *S. pombe* strains used in this study are listed in Table S3. Overexpression experiments were performed in minimal media (EMM) supplemented with uracil, adenine, and histidine at 28°C (Moreno et al., 1991). Cells were pre-cultured in EMM with supplemented with thiamine (5 µg/ml) at 28°C for 8 h, washed three times with EMM, and then transferred to thiamine-free EMM for grown prior to imaging. For plasmid construction, the GFP or NES-GFP fragments amplified from pTOW-h-NES-GFP (Kintaka et al., 2016) were subcloned into SalI/BamHI-digested pREP1. PCR primers for plasmid construction are listed in Table S4.

### Growth assays
Spotting assays were performed by growing cells to mid-log phase in minimal media and spotting onto EMM plates with or without thiamine. Plates were incubated at 28°C for 3 to 5 days. For liquid culture growth curves, cells were inoculated in EMM after the pre-culture in EMM supplemented with thiamine at 28°C. Cell number was measured at indicated time points using CellDrop (DeNovix).

### Microscopy and image analysis
Fluorescence imaging was carried out at 28°C using an Olympus IX83 inverted microscope system with UPLXAPO 100× objective lens (NA 1.45, immersion oil) and the ORCA-Fusion camera (Hamamatsu Photonics,

Japan). Images were captured in 0.3 or 0.4 µm z-sections over 5 µm using cellSens (Evident Scientific) for the IX83 system. Fluorescent intensity was measured using ImageJ (NIH). Time-lapse microscopy was performed using 35 mm Petri dish (MatTek corporation, USA) at 28°C, with image acquired at 5-min intervals. For DNA staining, DAPI and Hoechst 33342 were used for fixed and live cells, respectively. Cells were fixed by adding 50 µl of formaldehyde to 450 µl of culture. The frequency of cells with chromosome mis-segregation was determined by manually scoring cells exhibiting abnormalities similar to those shown in Fig. 1E. Cells exhibiting intranuclear microtubule bundles were identified based on the morphological features (e.g. microtubule confined within the nucleus or nuclear protrusions observed by the Cut11-mCherry signal) shown in Fig. S1B, and their frequency was quantified. For N/C ratio measurements, the volumes of cells and nuclei were calculated based on the manually measured values of the long and short axes of the cells (brightfield images) and nuclei (fluorescence images of Cut11-mCherry), using ImageJ as described previously (Newmann and Nurse, 2007).

## Statistical analysis

All experiments were performed independently in triplicate. Two-tailed unpaired Student's $t$-tests were used to assess differences in cell size and N/C ratio for each measurement between two groups. Chi-square tests were used to compare the frequency of cells with nuclear microtubule bundles, and, separately, the frequency of cells with chromosome mis-segregation.

## Acknowledgements

We thank P. Nurse for helpful discussions, H. Moriya for providing the plasmids, and D. Hirata for helpful discussions on this study. mCherry-Atb2 strain was provided by the National Bio-Resource Project (NBRP) – Yeast, Japan.

## Competing interests

The authors declare no competing or financial interests.

## Author contributions

Conceptualization: K.K.; Data curation: T.F.; Formal analysis: T.F., S.W., Y.I., K.K.; Funding acquisition: K.K.; Investigation: T.F., S.W., Y.I.; Supervision: K.K.; Validation: T.F.; Visualization: T.F.; Writing – original draft: K.K.; Writing – review & editing: M.M.

## Funding

This work was supported by the Japan Society for the Promotion of Science (JSPS KAKENHI) scientific research (B) grant number 24K01681, the Institute for Fermentation, Osaka (IFO) laboratory grant, and JST SPRING, grant number JPMJSP2132. Open Access funding provided by JSPS (24K01681). Deposited in PMC for immediate release.

## Data and resource availability

All relevant data and details of resources can be found within the article and its supplementary information.

## Peer review history

The peer review history is available online at https://journals.biologists.com/bio/lookup/doi/10.1242/bio.062331.reviewer-comments.pdf

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
