## [Peer Review File · Biology Open]

Nuclear enlargement induced by overexpression of nuclear export signal is associated with abnormal nuclear division in *Schizosaccharomyces pombe*

Takahiro Fujimoto, Suzu Watanabe, Yuko Imamura, Masaki Mizunuma and Kazunori Kume
DOI: 10.1242/bio.062331

Editor: Catherine L. Jackson

Review timeline

Submission to sister journal:	12 September 2025
Editorial decision at sister journal:	12 October 2025
Transfer to Biology Open:	24 October 2025
Accepted:	30 October 2025

Original submission to sister journal

First decision letter

MS TITLE: Nuclear size homeostasis ensures faithful chromosome segregation in *Schizosaccharomyces pombe*

AUTHORS: Kazunori Kume; Takahiro Fujimoto; Suzu Watanabe; Yuko Imamura; Masaki Mizunuma
ARTICLE TYPE: Short Report

We have now reached a decision on the above manuscript.

To see the reviewers' reports and a copy of this decision letter, please go to:

As you will see from their reports, the reviewers raise a number of substantial criticisms that prevent me from accepting your paper for publication.

I am very sorry to give you such disappointing news, but it takes a very enthusiastic recommendation by the referees for a manuscript to be accepted.

I do hope you find the comments of the reviewers helpful in allowing you to revise the manuscript for submission elsewhere, and many thanks for sending your work to us.

Reviewer 1

Comments from the Reviewers:

SUMMARY OF THE ADVANCE MADE IN THIS PAPER AND ITS POTENTIAL SIGNIFICANCE TO THE FIELD

Kume et al describe the phenotype that chromosome segregation errors occur in the NES-overexpression mutant, accompanied by an aberrant nuclear-to-cytoplasmic (N/C) volume ratio. They further demonstrate that these segregation errors are dependent on the accumulation of intranuclear microtubules. These findings are novel and highlight the biological significance of nuclear size in this field. Nevertheless, some aspects of the results and discussion are still limited in their current form. The following points should be considered before publication.

SUGGESTIONS TO AUTHORS

Major comments [Please request additional experiments only if they are essential for supporting the conclusions; authors should be encouraged to highlight any claims that are preliminary or speculative, or to discuss any pitfalls or alternative interpretations in a 'Limitations' section]

- Throughout the manuscript, the authors claim nuclear size homeostasis is essential for accurate chromosome segregation based on the experimental results obtained under conditions of abnormal N/C ratio. However, their current system may not be sufficient to support the general significance of the N/C ratio. In this study, abnormal N/C ratios were induced specifically through the ectopic accumulation of intranuclear microtubules. These data therefore support a link between nuclear size abnormality with intranuclear microtubules and chromosome segregation error, but not necessarily nuclear size abnormality in general. To strengthen the claim of general significance, it would be important to examine whether chromosome segregation errors are also observed in other mutants with abnormal N/C ratio, independent of intranuclear microtubule accumulation. If such data are not available, the author should consider moderating the conclusion and limiting the discussion to the context of N/C ratio changes associated with intranuclear microtubule accumulation.

- Some criteria used to define the cell division phenotypes are not clearly described in the Materials and Methods section. For readers unfamiliar with yeast research, it is difficult to evaluate whether the authors' determinations are appropriate. Specifically, further clarification is needed for:

1. Fidelity of chromosome segregation (Fig. 1E): The criteria for mis-segregation are particularly important. It remains unclear whether this category includes intranuclear chromosome deformation or abnormal DNA content caused by DNA replication errors.

2. Duration of forming spindle (Fig. 2C).

3. Definition of unequal cell division (Fig. 2C).

4. Measurement of N/C ratio (Fig. 3B).

- Regarding Fig.1, the authors state that cells overexpressing GFP or NES-GFP grew normally. To properly support this point, an additional control condition (yeast without genetic manipulation) should be included for comparison.

- In Fig.1D, statistical analysis is missing to support the claim of differences in the percentages between cells overexpressing GFP and GFP-NLS. Such statistical evaluation is necessary to substantiate this conclusion.

Minor comments

- The NES-GFP overexpression strain displays cells both with and without intranuclear microtubule protrusions. If the author could examine the correlation between the presence of these protrusions and chromosome segregation errors in each NES-GFP overexpression cell, their claim would be further emphasized. However, this analysis is not strictly required.

- In Fig. 2A, it would be preferable to set the time = 0 at the point of spindle formation, as in Figs. 2B-E.

- The meaning of "spindle-independent nuclear division" is unclear in the current version of the manuscript. Please provide a clear explanation.

- In Fig. 2C, it would be helpful to indicate spindle-independent nuclei with arrows.

- Is Fig. 2F derived from time-lapse imaging of the same yeast cell? If so, please include a time stamp. If not, the figure should ideally be replaced with time-lapse data, since such data are crucial to determine whether nuclei appear independently of spindle formation.

- It would be helpful to explicitly name each phenotype in Figs. 2C and 2D (e.g., "slow spindle formation" or "delayed spindle elongation"). For example, in Fig. 3A, the cells described as "in Fig.2C" and "in Fig.2D" are not identical to those in Fig.2, which may confuse.
- On line 107, the authors discuss "this transition may contribute to the increase in cell size observed in cells overexpressing NES-GFP (l.107)". The logic underlying this point is unclear and should be explained more explicitly.
- Please correct the reference description in the results section for Lemière et al., 2024.
- In several places, a space is missing between the number and unit (e.g., "18µm" should be "18 µm").

Reviewer 2

Comments for the author

SUMMARY OF THE ADVANCE MADE IN THIS PAPER AND ITS POTENTIAL SIGNIFICANCE TO THE FIELD

In this short report from Fujimoto et al., the authors investigate the consequences of overexpressing GFP fused to a nuclear export signal (NES-GFP) in fission yeast. They show that NES-GFP overexpression slows cell proliferation, and this cell cycle delay leads to larger cells. Mitotic spindles and nuclear divisions were abnormal in about half of the NES-GFP expressing cells, with spindle elongation persisting longer and some instances of nuclear division occurring in a spindle-independent manner. Spindle and nuclear division defects correlated with higher pre-division N/C ratios. Deletion of an importin-alpha gene (*imp1*) rescued some of the effects caused by NES-GFP overexpression while deletion of a gamma-tubulin complex linker gene (*mto2*) involved in cytoplasmic microtubule (MT) nucleation exacerbated some of the phenotypes. This study addresses an interesting and important problem, however I have concerns about how the authors are interpreting the results of their experiments as detailed below.

SUGGESTIONS TO AUTHORS

Major comments

1. The authors' main conclusion that nuclear size is important for faithful chromosome segregation is not well-substantiated. The authors propose that defects in mitotic progression and chromosome segregation are due to increased nuclear size resulting from NES-GFP expression. However, there are other interpretations of the data, perhaps more plausible, in which nuclear size is not the direct mediator of observed effects. For example, in the introduction the authors state that NES-GFP overexpression leads to nuclear accumulation of cargos, formation of intranuclear MT bundles, and nuclear enlargement. This statement is difficult to fully assess without knowing more about how NES-GFP is inducing nuclear enlargement, which appears to be described in another manuscript under review elsewhere. Nonetheless, it seems possible that NES-GFP overexpression is leading to nuclear accumulation of cargos that are the direct effectors of observed phenotypes. One model consistent with all of the data presented is that MT regulators accumulate in the nucleus and lead to the formation of aberrant intranuclear MT bundles during interphase (Fig. S1). This could very well lead to defects in regulation of the mitotic spindle (Fig. 2) which, in turn, could cause chromosome mis-segregation and cell cycle delay (Fig. 1). As a result of an extended G2, cell and nuclear sizes would increase (Fig. 1, 3, and S1), as was shown for various cell cycle mutants by Neumann and Nurse (PMID 17998401). Thus, NES-GFP could induce all of the observed effects through a mechanism that has nothing to do with nuclear size, and the observed increase in nuclear size is an indirect effect resulting from cell cycle delay.

The rescue experiment with *imp1* deletion is also not particularly convincing since *imp1* does not only affect nuclear size. Rescue could occur through a mechanism not involving nuclear size. For instance, the authors show that *imp1* deletion attenuated formation of intranuclear MTs induced by NES-GFP. Deletion of *imp1* could lead to reduced nuclear import of cargos that are accumulating due to NES-GFP overexpression. Reduced import of MT regulators could explain why intranuclear MT

bundles are no longer observed and this, in turn, would explain rescue of mitotic spindle defects, chromosome mis-segregation, and cell cycle delay, again with no direct involvement of nuclear size. The same logic would apply to *mto2* deletion that enhanced the formation of intranuclear MT bundles which could directly exacerbate observed phenotypes independently of effects on nuclear size. At least based on the data presented, the authors cannot conclude that nuclear size per se is functionally important for normal cell growth and chromosome segregation.

Another weakness is that the authors only use one approach to induce nuclear enlargement through NES-GFP overexpression. This further limits the strength of their conclusions. If the authors want to address the question of how nuclear size impacts mitotic fidelity and cell cycle progression, they should induce nuclear enlargement in multiple ways that act through distinct mechanisms. Along those lines, the authors state the *mto2* deletion should enhance nuclear expansion but that is not what they observed in the GFP control. Likewise, the link to nuclear size would be stronger if the authors investigated other mutants beyond *imp1* and *mto2* that suppress or enhance nuclear expansion in NES-GFP overexpressing cells.

2. The evidence for chromosome segregation defects caused by NES-GFP overexpression is weak (Figure 1E-F and 4B). Based on the DAPI images shown in Fig. 1E, it is not clear how the authors identified cells with chromosome segregation defects, and this is not described in the methods. If the authors are using the appearance of DAPI signal outside the nucleus as evidence of chromosome mis-segregation, this doesn't seem particularly convincing. This conclusion would be stronger if the authors used a complementary approach to quantify chromosome mis-segregation (e.g. minichromosome loss assay). A more minor issue is that Fig. 1F does not show data for the GFP control.

3. The formation of interphase intranuclear MT bundles induced by NES-GFP overexpression is not clear based on the images presented. In Figure S1A, I don't see much of a difference between GFP and NES-GFP overexpression in WT. It is not clear how cells with intranuclear MT bundles were quantified in Figure S1B as it is not described in the methods. In some cases, intranuclear MT bundles appear continuous with cytoplasmic MTs (e.g., 0 and 10 min in Fig. 2A, *mto2del* NES-GFP in Fig. S1A), so are the authors proposing these MT bundles pass through the nuclear envelope? In general, the localization of intranuclear MT bundles is difficult to visualize because both the nuclear envelope and MTs are labeled with the same mCherry fluorophore.

Minor comments

1. Fig. S1: there appears to be cell-to-cell variability in GFP and NES-GFP expression levels. Why? Do expression levels correlate with observed phenotypes? A good control would be to determine if GFP and NES-GFP expression levels are comparable in wild-type and in the *imp1* and *mto2* deletion mutants.

2. Does *imp1* deletion rescue the cell proliferation defect, cell cycle delay, and cell enlargement caused by NES-GFP overexpression? Does *mto2* deletion exacerbate these phenotypes?

3. It is curious that as a population the NES-GFP overexpressing cells show cell enlargement in Fig. 1C, but Table S1 indicates that only cells with the spindle phenotype shown in Fig. 2C were enlarged, accounting for only 23% of the cell population.

4. In the section titled "Genetic modulation of nuclear size alters the severity of mitotic phenotypes", the authors discuss percentages of mitotic cells with abnormal nuclear divisions, but in the corresponding Fig. 4B those numbers appear to refer to the percentage of mitotic cells with chromosome mis-segregation. The authors should clarify their language here. Do the *imp1* and *mto2* deletions affect the proportion of cells with abnormal nuclear divisions as defined in Fig. 2C-D?

5. Line 65: remove the word "increased"?

Transfer to Biology Open

Author response to reviewers' comments

Reviewer 1: SUMMARY OF THE ADVANCE MADE IN THIS PAPER AND ITS POTENTIAL SIGNIFICANCE TO THE FIELD

Kume et al describe the phenotype that chromosome segregation errors occur in the NES-overexpression mutant, accompanied by an aberrant nuclear-to-cytoplasmic (N/C) volume ratio. They further demonstrate that these segregation errors are dependent on the accumulation of intranuclear microtubules. These findings are novel and highlight the biological significance of nuclear size in this field. Nevertheless, some aspects of the results and discussion are still limited in their current form. The following points should be considered before publication.

SUGGESTIONS TO AUTHORS

Major comments [Please request additional experiments only if they are essential for supporting the conclusions; authors should be encouraged to highlight any claims that are preliminary or speculative, or to discuss any pitfalls or alternative interpretations in a 'Limitations' section]

- Throughout the manuscript, the authors claim nuclear size homeostasis is essential for accurate chromosome segregation based on the experimental results obtained under conditions of abnormal N/C ratio. However, their current system may not be sufficient to support the general significance of the N/C ratio. In this study, abnormal N/C ratios were induced specifically through the ectopic accumulation of intranuclear microtubules. These data therefore support a link between nuclear size abnormality with intranuclear microtubules and chromosome segregation error, but not necessarily nuclear size abnormality in general. To strengthen the claim of general significance, it would be important to examine whether chromosome segregation errors are also observed in other mutants with abnormal N/C ratio, independent of intranuclear microtubule accumulation. If such data are not available, the author should consider moderating the conclusion and limiting the discussion to the context of N/C ratio changes associated with intranuclear microtubule accumulation.

We have moderated the title, conclusion, and discussion to focus on the phenotypes observed in cells overexpressing NES-GFP, as this study specifically addresses N/C ratio changes associated with intranuclear microtubule bundles (Lines 2-3, 25-28, 61-71, 179-185, and 191-196).

- Some criteria used to define the cell division phenotypes are not clearly described in the Materials and Methods section. For readers unfamiliar with yeast research, it is difficult to evaluate whether the authors' determinations are appropriate. Specifically, further clarification is needed for:

1. Fidelity of chromosome segregation (Fig. 1E): The criteria for mis-segregation are particularly important. It remains unclear whether this category includes intranuclear chromosome deformation or abnormal DNA content caused by DNA replication errors.

We have added representative images showing chromosome mis-segregation in Fig.1E to clarify the criteria for this phenotype and described in the Materials and Methods section (Lines 241-243 and 396-399).

2. Duration of forming spindle (Fig. 2C).

We have added a detailed description of how the duration of spindle formation was measured in the figure legend of Fig.2C (Lines 413-414).

3. Definition of unequal cell division (Fig. 2C).

We have added a description of how "unequal nuclear division" was defined in the Results and Discussion section (Lines 119-120).

4. Measurement of N/C ratio (Fig. 3B).

We have added a detailed description of how the N/C ratios was measured in the Material and Methods section (Lines 246-249).

- Regarding Fig.1, the authors state that cells overexpressing GFP or NES-GFP grew normally. To properly support this point, an additional control condition (yeast without genetic manipulation) should be included for comparison.

Cells overexpressing GFP or NES-GFP were grown in the presence of thiamine, which represses gene expression under the *mt1* promoter. Therefore, the growth condition represents that of non-induced (wild type like) cells. We have added this explanation to the figure legend of Fig.1 (Lines 388-389).

- In Fig.1D, statistical analysis is missing to support the claim of differences in the percentages between cells overexpressing GFP and GFP-NLS. Such statistical evaluation is necessary to substantiate this conclusion.

We have performed statistical analysis for Fig.1D and added the results to the figure and legend (Lines 396-397).

Minor comments

- The NES-GFP overexpression strain displays cells both with and without intranuclear microtubule protrusions. If the author could examine the correlation between the presence of these protrusions and chromosome segregation errors in each NES-GFP overexpression cell, their claim would be further emphasized. However, this analysis is not strictly required.

We analyzed the time-lapse images of cells overexpressing NES-GFP and found that 50 % of interphase cells with intranuclear microtubule protrusions underwent normal nuclear division during mitosis (Extra table 1). Furthermore, the N/C ratio of these interphase cells was lower than 0.10 (Extra table 1). These results support our conclusion that abnormal nuclear size, rather than the presence of intranuclear microtubule protrusions itself, causes chromosome segregation errors.

Extra table 1 Frequency of normal and abnormal nuclear division in cells with intranuclear microtubule bundles, and N/C ratio of each interphase cell prior to mitotic entry (n=20, from time lapse images).

[NOTE: an image provided in confidence for the reviewers has been removed]

- In Fig. 2A, it would be preferable to set the time = 0 at the point of spindle formation, as in Figs. 2B-E.

As suggested, we have changed the time point in Fig. 2A so that time = 0 corresponds to the onset of spindle formation.

- The meaning of "spindle-independent nuclear division" is unclear in the current version of the manuscript. Please provide a clear explanation.

We have added a clear explanation of "spindle-independent nuclear division" in the Result section (Lines 119-120).

- In Fig. 2C, it would be helpful to indicate spindle-independent nuclei with arrows.

We have inserted arrows in Fig. 2C to indicate spindle-independent nuclei.

- Is Fig. 2F derived from time-lapse imaging of the same yeast cell? If so, please include a time stamp. If not, the figure should ideally be replaced with time-lapse data, since such data are crucial to determine whether nuclei appear independently of spindle formation.

No, the images in Fig.2F are snapshots, not from time-lapse imaging. We intended to show that the nuclei derived from spindle-independent division contain DNA, indicating that chromosome segregation occurred in these divisions. We also attempted time-lapse observation of DNA using Hoechst staining; however, the growth of Hoechst-stained cells was severely impaired, and thus

reliable time-lapse data could not be obtained.

- It would be helpful to explicitly name each phenotype in Figs. 2C and 2D (e.g., "slow spindle formation" or "delayed spindle elongation"). For example, in Fig. 3A, the cells described as "in Fig.2C" and "in Fig.2D" are not identical to those in Fig.2, which may confuse. We have named each phenotype in Figs. 2C and 2D as "unequal nuclear division" and "spindle-independent nuclear division", respectively (lines 119-123).

- On line 107, the authors discuss "this transition may contribute to the increase in cell size observed in cells overexpressing NES-GFP (l.107)". The logic underlying this point is unclear and should be explained more explicitly. As the logic of this description was unclear, we have deleted this sentence and now only present the result (lines 112-113).

- Please correct the reference description in the results section for Lemière et al., 2024. We have corrected the reference description for Lemière et al., 2024 (Line 42).

- In several places, a space is missing between the number and unit (e.g., "18µm" should be "18 µm"). We have added spaces between the numbers and units throughout the manuscript.

Reviewer 2: SUMMARY OF THE ADVANCE MADE IN THIS PAPER AND ITS POTENTIAL SIGNIFICANCE TO THE FIELD

In this short report from Fujimoto et al., the authors investigate the consequences of overexpressing GFP fused to a nuclear export signal (NES-GFP) in fission yeast. They show that NES-GFP overexpression slows cell proliferation, and this cell cycle delay leads to larger cells. Mitotic spindles and nuclear divisions were abnormal in about half of the NES-GFP expressing cells, with spindle elongation persisting longer and some instances of nuclear division occurring in a spindle-independent manner. Spindle and nuclear division defects correlated with higher pre-division N/C ratios. Deletion of an importin-alpha gene (*imp1*) rescued some of the effects caused by NES-GFP overexpression while deletion of a gamma-tubulin complex linker gene (*mto2*) involved in cytoplasmic microtubule (MT) nucleation exacerbated some of the phenotypes. This study addresses an interesting and important problem, however I have concerns about how the authors are interpreting the results of their experiments as detailed below.

SUGGESTIONS TO AUTHORS

Major comments

1. The authors' main conclusion that nuclear size is important for faithful chromosome segregation is not well-substantiated. The authors propose that defects in mitotic progression and chromosome segregation are due to increased nuclear size resulting from NES-GFP expression. However, there are other interpretations of the data, perhaps more plausible, in which nuclear size is not the direct mediator of observed effects. For example, in the introduction the authors state that NES-GFP overexpression leads to nuclear accumulation of cargos, formation of intranuclear MT bundles, and nuclear enlargement. This statement is difficult to fully assess without knowing more about how NES-GFP is inducing nuclear enlargement, which appears to be described in another manuscript under review elsewhere. Nonetheless, it seems possible that NES-GFP overexpression is leading to nuclear accumulation of cargos that are the direct effectors of observed phenotypes. One model consistent with all of the data presented is that MT regulators accumulate in the nucleus and lead to the formation of aberrant intranuclear MT bundles during interphase (Fig. S1). This could very well lead to defects in regulation of the mitotic spindle (Fig. 2) which, in turn, could cause chromosome mis-segregation and cell cycle delay (Fig. 1). As a result of an extended G2, cell and nuclear sizes would increase (Fig. 1, 3, and S1), as was shown for various cell cycle mutants by Neumann and Nurse (PMID 17998401). Thus, NES-GFP could induce all of the observed effects through a mechanism that has nothing to do with nuclear size, and the observed increase in

nuclear size is an indirect effect resulting from cell cycle delay.

We agree that NES-GFP overexpression may affect multiple nuclear processes and that the observed phenotypes could arise from combined effects, rather than from nuclear size increase alone. As described above, the formation of intranuclear microtubule bundles alone is unlikely to be the sole cause of chromosome segregation errors. To reflect this interpretation, we have moderated the title, conclusion and discussion, to focus on the phenotypes associated with NES-GFP overexpression, rather than on nuclear size as a general determinant (Lines 2-3, 25-28, 61-71, 179-185, and 191-196).

The rescue experiment with *imp1* deletion is also not particularly convincing since *imp1* does not only affect nuclear size. Rescue could occur through a mechanism not involving nuclear size. For instance, the authors show that *imp1* deletion attenuated formation of intranuclear MTs induced by NES-GFP. Deletion of *imp1* could lead to reduced nuclear import of cargos that are accumulating due to NES-GFP overexpression. Reduced import of MT regulators could explain why intranuclear MT bundles are no longer observed and this, in turn, would explain rescue of mitotic spindle defects, chromosome mis-segregation, and cell cycle delay, again with no direct involvement of nuclear size. The same logic would apply to *mto2* deletion that enhanced the formation of intranuclear MT bundles which could directly exacerbate observed phenotypes independently of effects on nuclear size. At least based on the data presented, the authors cannot conclude that nuclear size per se is functionally important for normal cell growth and chromosome segregation. We acknowledge that the rescue effects observed in the *imp1* deletion and *mto2* deletion backgrounds could involve mechanisms other than nuclear size regulation, such as altered nuclear import or microtubule organization. In the revised manuscript, we have added detailed descriptions indicating that *imp1* deletion reduced, whereas *mto2* deletion enhanced, the formation of intranuclear bundle associated with nuclear enlargement, and that these genetic backgrounds altered the frequency of abnormal nuclear divisions (Lines 152-156, 158-165, and 168-185). These results collectively support the idea that nuclear enlargement, possibly in combination with intranuclear microtubule formation, contributes to chromosome segregation defects. Accordingly, we have moderated the title, conclusion, and discussion to limit our claims to the context of NES-GFP-induced phenotypes (Lines 2-3, 25-28, 61-71, 179-185, and 191-196).

Another weakness is that the authors only use one approach to induce nuclear enlargement through NES-GFP overexpression. This further limits the strength of their conclusions. If the authors want to address the question of how nuclear size impacts mitotic fidelity and cell cycle progression, they should induce nuclear enlargement in multiple ways that act through distinct mechanisms. Along those lines, the authors state the *mto2* deletion should enhance nuclear expansion but that is not what they observed in the GFP control. Likewise, the link to nuclear size would be stronger if the authors investigated other mutants beyond *imp1* and *mto2* that suppress or enhance nuclear expansion in NES-GFP overexpressing cells.

As mentioned above, this study focused on the phenotypes associated with NES-GFP overexpression. Accordingly, we have moderated the title, conclusion and discussion to limit our claims to this specific context (Lines 2-3, 25-28, 61-71, 179-185, and 191-196).

2. The evidence for chromosome segregation defects caused by NES-GFP overexpression is weak (Figure 1E-F and 4B). Based on the DAPI images shown in Fig. 1E, it is not clear how the authors identified cells with chromosome segregation defects, and this is not described in the methods. If the authors are using the appearance of DAPI signal outside the nucleus as evidence of chromosome mis-segregation, this doesn't seem particularly convincing. This conclusion would be stronger if the authors used a complementary approach to quantify chromosome mis-segregation (e.g. minichromosome loss assay). A more minor issue is that Fig. 1F does not show data for the GFP control.

We have clarified in the Material and methods section (Lines 241-243) how chromosome segregation defects were identified. Representative images of cells showing chromosome mis-segregation have been included in Fig. 1E, and the GFP control has been added in the same figure. In Figure 4B, the frequency of abnormal chromosome segregation was quantified based on DNA signals remaining within the nuclear envelope (Cut11-mCherry) after division, as observed by Hoechst staining in living cells. These results consistently support that overexpression of NES-GFP

induces chromosome segregation defects.

3. The formation of interphase intranuclear MT bundles induced by NES-GFP overexpression is not clear based on the images presented. In Figure S1A, I don't see much of a difference between GFP and NES-GFP overexpression in WT. It is not clear how cells with intranuclear MT bundles were quantified in Figure S1B as it is not described in the methods. In some cases, intranuclear MT bundles appear continuous with cytoplasmic MTs (e.g., 0 and 10 min in Fig. 2A, *mto2del* NES-GFP in Fig. S1A), so are the authors proposing these MT bundles pass through the nuclear envelope? In general, the localization of intranuclear MT bundles is difficult to visualize because both the nuclear envelope and MTs are labeled with the same mCherry fluorophore.

We have replaced the images in Fig. S1 to more clearly show NES-GFP overexpressing cells containing intranuclear microtubule bundles. In addition, we have added middle-section images of Cut11-mCherry and GFP-LacI-NLS, which demonstrate that the formation of intranuclear microtubule bundles is accompanied by a morphological change in nuclear shape from spherical to lemon-like. Based on these images, we quantified the proportion of cells containing intranuclear microtubule bundles, as described in the Material and Methods section (Lines 243-246).

Minor comments

1. Fig. S1: there appears to be cell-to-cell variability in GFP and NES-GFP expression levels. Why? Do expression levels correlate with observed phenotypes? A good control would be to determine if GFP and NES-GFP expression levels are comparable in wild-type and in the *imp1* and *mto2* deletion mutants.

In fission yeast at 28°C, cells under the control of the *nmt1* promoter begin to show overexpression after approximately 16 hours of induction, and the proportion of strongly expressing cells gradually increases over the following several hours (extra fig.1). Consequently, cell-to-cell variability in expression levels is commonly observed in this system. We found that NES-GFP expression level correlated with the observed phenotypes: cells with higher expression levels exhibited a greater increase in the N/C ratio.

[NOTE: an image provided in confidence for the reviewers has been removed]

Extra Fig.1 Average fluorescence intensity of NES-GFP at the indicated time points (n = 30). Cells were grown in EMM medium at 28°C.

2. Does *imp1* deletion rescue the cell proliferation defect, cell cycle delay, and cell enlargement caused by NES-GFP overexpression? Does *mto2* deletion exacerbate these phenotypes?

No, the deletion of *imp1* did not rescue the cell proliferation defect caused by NES-GFP overexpression (Extra Fig. 2A). However, according to interphase cell volume (Table S1), cell enlargement was partially suppressed at 16 - 20 hours. In contrast, *mto2* deletion enhanced cell enlargement at 16 hours, when the formation of intranuclear microtubule bundles and nuclear enlargement induced by NES-GFP overexpression were promoted. Regarding the cell cycle delay observed NES-GFP overexpressed cells, this phenotype was subtle and therefore difficult to evaluate. Indeed, we didn't observe any differences in the frequency of septated cells among the WT, *imp1* deletion and *mto2* deletion strains overexpressing NES-GFP (Extra Fig. 2B).

[NOTE: an image provided in confidence for the reviewers has been removed]

Extra Fig.2 (A) Growth properties of WT and *imp1*Δ cells overexpressing GFP or NES-GFP and (B) the frequency of septated cells in WT, *imp1*Δ, and *mto2*Δ cells overexpressing GFP or NES-GFP at indicated time points. Cells were grown in EMM medium at 28°C.

3. It is curious that as a population the NES-GFP overexpressing cells show cell enlargement in Fig. 1C, but Table S1 indicates that only cells with the spindle phenotype shown in Fig. 2C were enlarged, accounting for only 23% of the cell population.

We agree that this point is intriguing. We would like to further investigate the relationship between cell size and the spindle phenotype in future study.

4. In the section titled "Genetic modulation of nuclear size alters the severity of mitotic phenotypes", the authors discuss percentages of mitotic cells with abnormal nuclear divisions, but in the corresponding Fig. 4B those numbers appear to refer to the percentage of mitotic cells with chromosome mis-segregation. The authors should clarify their language here. Do the *imp1* and *mto2* deletions affect the proportion of cells with abnormal nuclear divisions as defined in Fig. 2C-D?

As Fig.4B shows the frequency of cells with abnormal nuclear division, we clarified this point in Fig.4B and manuscript (Lines 158-165 and 173-179).

5. Line 65: remove the word "increased"?
We removed the word "increased" (Line 71).

6. Table S1 refers to Fig. 3C but there is no panel C.
We changed Fig. 3C to Fig. 3B in Table S1.

7. While perhaps outside the scope of this manuscript, a further exploration of the mechanism of action of NES-GFP overexpression would be useful. If NES-GFP is overloading the nuclear export machinery, would Crm1 overexpression rescue?

Yes, Crm1 overexpression partially rescued the nuclear size increase induced by NES-GFP overexpression at 18 hours. However, since Crm1 overexpression alone caused severe growth defects, we could not assess whether it also suppressed the growth defect phenotype observed in NES-GFP overexpression cells.

[NOTE: an image provided in confidence for the reviewers has been removed]

Extra Fig.3 Growth properties (A) and N/C ratios (B) of WT cells overexpressing GFP or NES-GFP, and WT cells co-overexpressing GFP and Crm1 or NES-GFP and Crm1. Cells were grown in EMM medium at 28°C.

Second decision letter

MS ID#: bio. bio.062331

MS Title: Nuclear enlargement induced by overexpression of nuclear export signal is associated with abnormal nuclear division in *Schizosaccharomyces pombe*

Authors: Takahiro Fujimoto; Suzu Watanabe; Yuko Imamura; Masaki Mizunuma; Kazunori Kume
Article Type: Research Article

After careful consideration of your manuscript, and the revisions you have carried out in response to reviews of a previously submitted version, I am happy to tell you that your manuscript has been accepted for publication in Biology Open, pending our standard publication integrity checks. It was accepted on 30th October 2025.